# SLASHED NORMAL: PARAMETERIZE NORMAL POSTERIOR DISTRIBUTIONS WITH KL AMPLITUDE

## ABSTRACT

We present Slashed Normal, a novel parameterization for the normal posterior distribution in variational-inference-based latent variable models. Slashed Normal takes a simple form resembling conventional practice, but uses the new stdplus activation function to derive the standard deviation instead of softplus or exp. Although taking this simple form, the Slashed Normal establishes a direct connection between the squared $L^2$-norm of the raw neural network output, termed *KL amplitude*, and the exact KL divergence value between the prior and the posterior. As a result, this parameterization enables a direct control of the KL divergence value, which is usually interpreted as the rate from the rate-distortion perspective for variational autoencoders. We demonstrate the versatility of Slashed Normal through theoretical analysis and experiments, showcasing its ability to provide good insight about the posterior distribution, explicit control over the KL divergence, and mitigate posterior collapse.

## 1 INTRODUCTION

Variational inference-based latent variable models, particularly Variational Autoencoders (VAEs) (Kingma and Welling, 2013; Higgins et al., 2016), have become fundamental tools in stochastic modeling with deep neural networks. At the core of VAE training lies a crucial balance between reconstruction and regularization. The regularization term, expressed as the Kullback-Leibler (KL) divergence between the posterior and prior of the latent variable, plays a pivotal role in shaping the model's behavior. This KL divergence, often interpreted as the model's *rate*, quantifies the information encoded in latent variables and significantly influences the quality of learned representations.

However, the promise of VAEs is tempered by persistent challenges that have affected researchers and practitioners alike, such as numerical instability (Vahdat and Kautz, 2020; Child, 2021) and posterior collapse (Bowman et al., 2015; Razavi et al., 2019; Lucas et al., 2019; Dai et al., 2019). Numerical instability manifests as large spikes in training loss, while posterior collapse results in the model ignoring a substantial portion of latent codes, hindering the learnability of the latent-variable model. These issues have been partially attributed to the KL divergence term in those individual works, motivating the need to obtain control over this component.

Moreover, various applications require direct manipulation of KL values. For instance, disentangled representation learning (Higgins et al., 2016) relies on careful control of KL divergence to achieve interpretable latent spaces. Prediction attribution methods (Jiang et al., 2020; Schulz et al., 2020) use KL divergence to quantify information flow. Data compression techniques (Ballé et al., 2018; Huang et al., 2020; Flamich et al., 2020) directly relate KL divergence to encoding length. In these scenarios, precise control over KL divergence is not just beneficial but essential for achieving desired outcomes.

Existing methods for controlling KL divergence often rely on indirect mechanisms, such as adjusting the weight $\beta$ of the KL term in the loss function. However, this approach can lead to tuning difficulties and potential instabilities during training. To illustrate this challenge, we present a motivational example in Figure 1.

Figure 1 compares $\beta$-tuning with direct rate control, as enabled by the proposed parameterization, in a Variational Information Bottleneck (VIB, Alemi et al. (2017)) context. The top panel shows that when tuning $\beta$, a sharp accuracy drop (to 0.5) occurs at a threshold $\beta_0$ [1], beyond which all latents

---

[1]In this toy example, $\beta_0$ is known as a function of the label flipping probability

collapse (Wu et al., 2020). Optimal performance is precariously close to this threshold. In contrast, the bottom panel demonstrates that directly tuning the rate yields more stable performance across a range of reasonable values. Achieving certain optimal rates via $\beta$-tuning requires carefully designed schedules, with most popular KL warmup schedules failing except for adaptive controllers like GECO (Rezende and Viola, 2018).

To address these challenges, we propose *Slashed Normal*, a novel parameterization of the posterior Normal distribution relative to a specified Gaussian prior. Our approach offers several key advantages:

- **Direct KL Control**: Slashed Normal establishes a direct link between the squared $L^2$-norm of the raw network output and the KL divergence, allowing direct control of the channel capacity in latent codes.
- **Simplicity**: The parameterization closely resembles conventional VAE practices, facilitating easy adoption.
- **Theoretical insights**: Our formulation provides new perspectives on phenomena like posterior collapse. Due to the resemblance between our parameterization and conventional parameterization, we argue that our results also approximately hold for the conventional parameterization, especially for those using *softplus* activation.
- **Unification**: Slashed Normal generalizes several existing KL control techniques for mitigating posterior collapse for Gaussian VAEs under a single framework.
- **New capabilities**: It enables novel approaches such as fixed-rate variational information bottlenecks.

This paper focuses on the theoretical construction, mathematical properties, and initial demonstrations of Slashed Normal in addressing key challenges in variational inference. Our work not only offers a powerful new tool for variational inference but also deepens our understanding of the role of KL divergence in latent variable models.

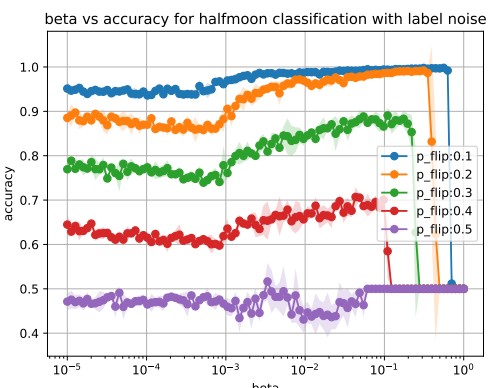

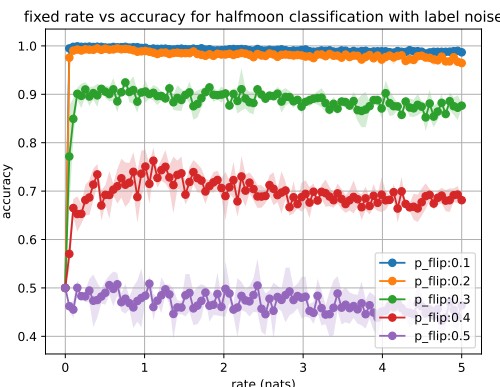

Figure 1: Half moon classification with random label noise. $\beta$-tuning (Top) vs. rate tuning (Bottom).

While we provide initial experimental results to validate our theoretical findings, exhaustive empirical comparisons across all possible applications are beyond the scope of this initial work. Our primary goal is to introduce Slashed Normal as a novel tool for the variational inference toolkit, laying the groundwork for future research and applications.

## 2 BACKGROUND

### 2.1 VARIATIONAL AUTOENCODERS

Variational Autoencoders (VAEs) (Kingma and Welling, 2013) model the data generation process as $\mathbf{z} \sim p(\mathbf{z}), \mathbf{x} \sim p_\theta(\mathbf{x}|\mathbf{z})$, where $p(\mathbf{z})$ is the prior distribution of the latent variable $\mathbf{z}$, and $p_\theta(\mathbf{x}|\mathbf{z})$ is the decoder that generates data $\mathbf{x}$ from $\mathbf{z}$. The encoder $q_\phi(\mathbf{z}|\mathbf{x})$ approximates the true posterior $p(\mathbf{z}|\mathbf{x})$. The VAE training objective is:

$$\mathcal{L}(\phi, \theta) = \mathbb{E}_{p_{\text{data}}(\mathbf{x})}\Big\{\underbrace{\mathbb{E}_{q_\phi(\mathbf{z}|\mathbf{x})}[-\log p_\theta(\mathbf{x}|\mathbf{z})]}_{\text{Reconstruction/Distortion}} + \underbrace{\beta D_{KL}(q_\phi(\mathbf{z}|\mathbf{x})\|p(\mathbf{z}))}_{\text{Regularization/Rate}}\Big\}, \tag{1}$$

where $p_{\text{data}}(\mathbf{x})$ is the empirical data distribution, $D_{KL}(q_\phi(\mathbf{z}|\mathbf{x})||p(\mathbf{z}))$ denotes the KL divergence between the variational approximation $q_\phi(\mathbf{z}|\mathbf{x})$ and the prior $p(\mathbf{z})$. The parameter $\beta$, introduced in (Higgins et al., 2016), controls the regularization strength. From a compression perspective, these terms are sometimes referred to as *distortion* and *rate* (Park et al., 2020), and $\beta$ governs the strength of compression.

In our work, we focus on the most common case where both prior and posterior are Gaussian distributions.

## 2.2 Posterior Collapse

Unfortunately, VAE training often suffers from *posterior collapse*, a phenomenon where posterior distributions become indistinguishable from the prior, rendering latent variables uninformative about the data. The phenomenon of posterior collapse could be attributed to model convergence to spurious local optima (Lucas et al., 2019; Dai et al., 2019) or poor global optima (Yacoby et al., 2020) that can explain data equally well as the good global optimum.

Mitigation strategies include clipping the KL divergence loss term (Kingma et al., 2014), enforcing a parameterization with a lower bound on the KL divergence (Davidson et al., 2018; Razavi et al., 2019; Zhu et al., 2020), scheduling or adaptively controlling the KL weight $\beta$ (Bowman et al., 2015; Fu et al., 2019; Shao et al., 2020; Rezende and Viola, 2018), limiting the decoder capacity (Bowman et al., 2015; Rey, 2021), enforcing specific properties in the network architecture (Wang et al., 2021; Kinoshita et al., 2023), and exploring less affected network architectures (Child, 2021).

## 2.3 Deep Variational Information Bottleneck

The Deep Variational Information Bottleneck (DVIB) (Alemi et al., 2017) generalizes VAEs beyond autoencoding. It uses $p(\mathbf{y}|\mathbf{z})p(\mathbf{z})/q(\mathbf{z}|\mathbf{x})$ to predict target $\mathbf{y}$ from input $\mathbf{x}$, learning a compressed representation that preserves prediction-relevant information. DVIB has shown effectiveness in neural network regularization, adversarial robustness (Alemi et al., 2017), and low-resource fine-tuning of large language models (mahabadi et al., 2021).

## 2.4 Residual Normal Distribution

The concept of parameterizing posterior distributions relative to the prior distribution has been previously explored in (Vahdat and Kautz, 2020). In their work, the posterior distribution, termed the *Residual Normal Distribution*, is expressed in terms of the relative mean $\Delta\mu$ and the relative standard deviation $\Delta\sigma$ with respect to the mean $\mu_0$ and standard deviation $\sigma_0$ of the prior Gaussian. This parameterization aims to facilitate training and is formulated as follows in the univariate case:

$$\mu = \mu_0 + \Delta\mu, \sigma = \sigma_0 \Delta\sigma. \tag{2}$$

The KL divergence term in their parameterization is computed as

$$D_{\text{KL}}(\mathcal{N}(\mu, \sigma^2)||\mathcal{N}(\mu_0, \sigma_0)) = \frac{1}{2}\left(\frac{\Delta\mu^2}{\sigma_0^2} + \Delta\sigma^2 - \log\Delta\sigma^2 - 1\right). \tag{3}$$

Our work extends this concept, deriving a parameterization where KL divergence depends solely on relative parameters, enabling explicit modeling of the KL divergence.

# 3 Slashed Normal: *KL Amplitude* Parameterized Gaussian Distribution

In this section, we introduce *Slashed Normal*, a novel parameterization for the Gaussian posterior that is relative to a specified Gaussian prior. Motivated by the need for direct control over KL divergence in variational inference, as discussed in the introduction, the derivation starts from an attempt to incorporate the KL divergence quantity as one parameter of the posterior distribution. This approach leads to a simple yet powerful parameterization that offers explicit control over the exact value of the KL divergence.

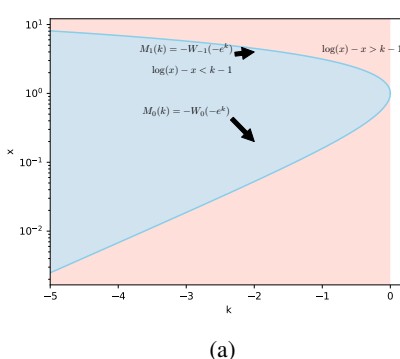
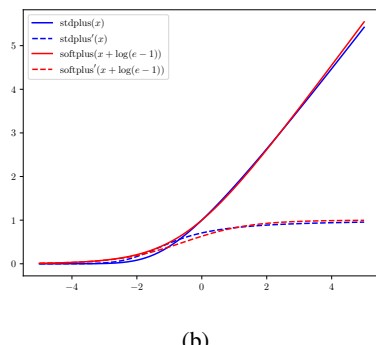

(a)                                                                                      (b)

Figure 2: *(a)* Upper and lower roots of Eq. (6) when $k \leq 0$, shown as the intersection between the two colored regions. *(b)* The proposed *stdplus* function and its derivative. We can see that this function resembles the 1-centered *softplus* function.

We will now derive this parameterization step by step, beginning with the basic parameter constraints and progressing to a general formulation applicable to multivariate Gaussian distributions.

### 3.1 THE PARAMETER CONSTRAINTS

For investigating how to incorporate the KL divergence quantity as one parameter of the posterior distribution, we begin with the analytic expression for the KL divergence between a univariate normal distribution $\mathcal{N}(\mu, \sigma^2)$ and a standard normal distribution $\mathcal{N}(0, 1)$:

$$D_{KL}(\mathcal{N}(\mu, \sigma^2)||\mathcal{N}(0,1)) = \frac{1}{2}\left(\mu^2 + \sigma^2 - \log(\sigma^2) - 1\right). \tag{4}$$

Let $D_{KL}(\mathcal{N}(\mu, \sigma^2)||\mathcal{N}(0,1)) = \delta$, we have

$$\log(\sigma^2) - \sigma^2 = -1 - (2\delta - \mu^2). \tag{5}$$

Denoting $k = -(2\delta - \mu^2)$ and $x = \sigma^2$, we arrive at:

$$\log(x) - x = k - 1. \tag{6}$$

Taking exponential on both sides, we obtain

$$xe^{-x} = e^{k-1} \implies (-x)e^{(-x)} = (-e^{k-1}), \tag{7}$$

which has the form of $ye^y = z$. The solution to this equation is given by the Lambert W function (Corless et al., 1996): $y = W(z)$.

Figure 2a illustrates the solutions to Eq. (6). When $k \leq 0$, real roots exist. These roots, named $x = M_0(k)$ and $x = M_1(k)$, can be directly represented using the two real branches of the Lambert W function:

$$M_0(k) = -W_0(-e^{k-1})$$
$$M_1(k) = -W_{-1}(-e^{k-1}) \tag{8}$$

Substituting $x$ and $k$ with the original variables, we have:

$$\sigma^2 = M_{\{0,1\}}(-(2\delta - \mu^2)), \tag{9}$$

where $M_{\{0,1\}}$ denotes either $M_0$ or $M_1$. We can easily verify that $\mathcal{D}_{KL}(\mathcal{N}(\mu, \sigma^2)||\mathcal{N}(0,1)) = \delta$.

While the resulting parameterization $(\mu, \delta)$ achieves our goal of incorporating $\delta$ as a parameter, it has two significant drawbacks: 1) it can only represent one branch of variances (either $M_0$ or $M_1$); 2) the derivative of the variance with respect to $\delta$, i.e., $\frac{\partial \sigma^2}{\partial \delta} = -2\frac{\partial x}{\partial k}$ goes to infinity as $\delta$ approaches 0 (see Fig. 2a). These limitations motivate the development of a more robust parameterization, which we introduce in the next subsection.

## 3.2 THE *KL Amplitude* PARAMETERIZATION

Examining Eq. 9, we see that if we define variables $a = \mu/\sqrt{2}$, and $b = \pm\sqrt{\delta - \mu^2/2}$, then KL divergence $\delta$ can be expressed as:
$$\delta = a^2 + b^2. \tag{10}$$

Substituting $(\mu, \delta)$ in Eq. 9 with $(a, b)$, we derive a new way to parameterize the normal distribution $\mathcal{N}(\mu, \sigma^2)$:
$$\mu = \sqrt{2}a,$$
$$\sigma^2 = M_{\{0,1\}}(-2b^2)). \tag{11}$$

In this parameterization, $\mu$ is controlled by $a$ while $\sigma^2$ is controlled by $b$, and the KL divergence equals the sum of squares of $a$ and $b$. Inspired by the concept of *probability amplitude* in quantum physics, we combine these parameters into a complex number $\psi$:
$$\psi = a + bi. \tag{12}$$

This complex number[2] combines the raw parameters for both mean and variance. We term $\psi$ the *KL amplitude*, as its squared modulus directly represents the KL divergence: $\delta = |\psi|^2$.

With a signed imaginary part, the two branches of $M(\cdot)$ can be further unified into a single function, using the sign of $b$ to select which branch to use. Additionally, for convenience, we make the designated function that glues the two branches to compute the standard deviation instead of the variance. This function, which we call *stdplus*, is defined as follows:
$$stdplus(x) = \begin{cases} \sqrt{M_0(-x^2)}, x < 0 \\ \sqrt{M_1(-x^2)}, x >= 0 \end{cases}. \tag{13}$$

This leads to our final parameterization, which we call Slashed Normal $\mathcal{\tilde{N}}(\psi)$:
$$\mathcal{N}(\mu, \sigma^2) = \mathcal{\tilde{N}}(\psi),$$
$$s.t. \quad \psi = a + bi,$$
$$\mu = \sqrt{2}a,$$
$$\sigma = stdplus(\sqrt{2}b). \tag{14}$$

The *stdplus* function and its derivative, depicted in Fig. 2b, resembles those of the 1-centered *softplus* function, which is shifted from the original *softplus* function such that it achieves 1 at $x = 0$. It can serve as a direct replacement for softplus or exp in computing standard deviations. More details about *stdplus* function, including the numerical recipe, is given in appendix A.

This complex-valued parameterization maintains explicit control over the KL divergence while addressing the unbounded derivative issue encountered in Section 3.1. It also closely resembles conventional VAE parameterizations, facilitating easy adoption in existing models.

## 3.3 GENERALIZE TO GENERAL UNIVARIATE GAUSSIAN PRIORS

We can extend the Slashed Normal parameterization to be relative to a univariate Gaussian with mean $\mu_0$ and variance $\sigma_0^2$:
$$\mathcal{N}(\mu, \sigma^2) = \mathcal{\tilde{N}}(\psi, \mu_0, \sigma_0^2)$$
$$s.t. \quad \psi = a + bi$$
$$\mu = \mu_0 + \sqrt{2}\sigma_0 a$$
$$\sigma = \sqrt{\sigma_0^2 M_{\{0,1\}}(-(2\delta - \frac{(\mu - \mu_0)^2}{\sigma_0^2}))},$$
$$= \sigma_0 stdplus(\sqrt{2}b). \tag{15}$$

This parameterization maintains the key property:
$$D_{KL}(\mathcal{N}(\mu, \sigma^2)||\mathcal{N}(\mu_0, \sigma_0^2)) = |\psi|^2. \tag{16}$$

---

[2]One may argue that the use of complex numbers is not necessary, however we identify that complex numbers are conceptually simpler among other alternative equivalent forms.

### 3.4 GENERALIZE TO MULTIVARIATE GAUSSIAN DISTRIBUTIONS

We can further extend the parameterization to multivariate Gaussian distributions with full covariance matrices for both the prior $\mathcal{N}(\boldsymbol{\mu}_0, \boldsymbol{\Sigma}_0)$ and posterior $\mathcal{N}(\boldsymbol{\mu}, \boldsymbol{\Sigma})$:

$$
\begin{aligned}
\mathcal{N}(\boldsymbol{\mu}, \boldsymbol{\Sigma}) =& \mathscr{N}(\boldsymbol{\psi}, \boldsymbol{P}, \boldsymbol{\mu}_0, \boldsymbol{\Sigma}_0) \\
s.t. \quad & \boldsymbol{\psi} = \boldsymbol{a} + \boldsymbol{b}i \\
& \boldsymbol{\mu} = \boldsymbol{\mu}_0 + \sqrt{2}\boldsymbol{\Sigma}_0^{1/2}\boldsymbol{a}, \\
& \boldsymbol{\Sigma}^{1/2} = \boldsymbol{\Sigma}_0^{1/2}\boldsymbol{P}\mathrm{diag}(stdplus(\sqrt{2}\boldsymbol{b})),
\end{aligned}
\tag{17}
$$

where complex vector $\boldsymbol{\psi} = \boldsymbol{a} + \boldsymbol{b}i$, $\boldsymbol{P}$ is an orthogonal matrix, and $\boldsymbol{\Sigma}_0^{1/2}$ is a matrix such that $\boldsymbol{\Sigma}_0 = (\boldsymbol{\Sigma}_0^{1/2})(\boldsymbol{\Sigma}_0^{1/2})^T$. This generalization comes from factorization of the covariance matrix.

This parameterization maintains the property:

$$
D_{KL}(\mathscr{N}(\boldsymbol{\psi}, \boldsymbol{P}, \boldsymbol{\mu}_0, \boldsymbol{\Sigma}_0)||\mathcal{N}(\boldsymbol{\mu}_0, \boldsymbol{\Sigma}_0)) = \boldsymbol{\psi}^H\boldsymbol{\psi}.
\tag{18}
$$

Notably, this generalization is applicable to priors that are degenerate multivariate normal distributions. In such cases, both the prior and posterior have support over an affine subspace of $\mathbb{R}^k$: $\{\boldsymbol{\mu}_0 + \boldsymbol{\Sigma}_0^{1/2}\boldsymbol{z} : \boldsymbol{z} \in \mathbb{R}^k\}$, where $k$ is the dimension of the vector; the matrix $\boldsymbol{\Sigma}_0^{1/2}$ of the prior is not required to be positive definite. A detailed derivation of this multivariate case is provided in appendix B.

## 4 BENEFITS OF SLASHED NORMAL

In this section, we demonstrate the practical advantages of the Slashed Normal parameterization.

### 4.1 VARIATIONAL AUTOENCODER WITH SLASHED NORMAL

As a concrete example, we demonstrate how the proposed Slashed Normal can simplify the formulation of a variational autoencoder with diagonal Gaussian latents.

Let $\boldsymbol{\psi}(\mathbf{x}) : \mathbb{R}^{N_1} \to \mathbb{C}^{N_2}$ be an encoder that maps from the data space to the KL amplitude latent space, where $N_1$ is the data dimension and $N_2$ is the latent dimension. Using Slashed Normal, we can express the evidence lower bound (ELBO) loss for a vanilla VAE as:

$$
\mathcal{L} = \mathop{\mathbb{E}}_{\mathbf{x}\sim p_{\mathrm{data}}(\mathbf{x})} \left\{ \underbrace{\mathop{\mathbb{E}}_{\mathbf{z}\sim\mathscr{N}(\mathbf{z};\boldsymbol{\psi}(\mathbf{x}))}[-\log p(\mathbf{x}|\mathbf{z})]}_{\text{Reconstruction}} + \underbrace{\boldsymbol{\psi}^H(\mathbf{x})\boldsymbol{\psi}(\mathbf{x})}_{\text{KL divergence}} \right\}.
\tag{19}
$$

Remarkably, the KL divergence term now exclusively comprises the squared $L^2$-norm of the raw encoder output $\boldsymbol{\psi}(\mathbf{x})$. Consequently, the entire objective takes the form of a $L^2$ regularized autoencoder with a stochastic reconstruction loss. Notably, this formulation eliminates all potentially unstable operations, e.g., log/exp, which previously requires clipping the range of the input to prevent numerical problems. This property likely improves the numerical stability of training.

### 4.2 EXPLICIT CONTROL OF KL DIVERGENCE

Explicit control, either through inequality or equality constraints, of the KL divergence (rate) term can be directly achieved by manipulating the $L^2$-norm of $\boldsymbol{\psi}(\mathbf{x})$, that is, the KL amplitude as a function of the input . Denoting $\tilde{\boldsymbol{\psi}}(\mathbf{x})$ as the raw neural network output, controlling the KL divergence value can be accomplished as follows:

$$
\boldsymbol{\psi}(\mathbf{x}) = \delta^{1/2}(\mathbf{x})\mathbf{normalize}(\tilde{\boldsymbol{\psi}}(\mathbf{x}))
\tag{20}
$$

Here $\delta^{1/2}(\mathbf{x})$ is the squared root rate function, which can a function of each input, or a constant for all inputs. The function $\mathbf{normalize}(\cdot)$ normalizes the input to unit $L^2$-norm.

This renormalization is equivalent to fixing the channel capacity, as demonstrated by the following theorem:

**Theorem 4.1.** *For* $\mathbf{z} \sim \mathscr{N}(\boldsymbol{\psi}(\mathbf{x}))$, *we have*

$$I(X; Z) \leq \mathbb{E}_{\mathbf{x}}\left[D_{KL}(q(\mathbf{z}|\mathbf{x})||p(\mathbf{z}))\right] = \mathbb{E}_{\boldsymbol{x}}||\boldsymbol{\psi}(\mathbf{x})||_2^2 = \textit{Channel Capacity}, \tag{21}$$

*where the equality is achieved when* $D_{KL}(q(\mathbf{z})||p(\mathbf{z})) = 0$.

*Proof.* See appendix C. $\qquad\square$

From this perspective, the stochastic layer defined by $\mathbf{z} \sim \mathscr{N}(\boldsymbol{\psi}(\mathbf{x}))$ can be viewed as a neural network component that imposes a predefined channel capacity, which functions similarly as *Gaussian Dropout* (Rey and Mnih, 2021), but with manageable channel capacity.

We then identify that different normalization schemes carry distinct information-theoretic implications. Assume that the raw network output $\tilde{\boldsymbol{\psi}}(\mathbf{x})$ for a minibatch has the shape $N \times K$, where $N$ and $K$ denote the batch size (*batch*) and the dimensionality of $\tilde{\boldsymbol{\psi}}$ (*dimension*), respectively, and a global squared root rate function $\delta^{1/2}(\mathbf{x}) = \delta^{1/2}$ is used. Then for the following normalization options:

1. *Batch*: normalize jointly across (batch, dimension): In this case, the total rate for the batch is $\delta$, The (average) rate per instance can be approximated as $\frac{\delta}{N}$.

2. *Instance*: normalize across (dimension): In this case, each instance in a mini-batch is forced to have a total rate of $\delta$.

3. *Feature*: normalize across (batch): In this case, every dimension of $\psi$ must have a total rate of $\delta$ over the batch. It corresponds to the case where all latent dimensions are forced to be active and have an average rate of $\frac{\delta}{N}$ per instances. This strategy can be viewed as a generalization of (Zhu et al., 2020), which directly applies the batchnorm to posterior means, together with a fixed scale parameter to enforce a lower bound on KL divergence.

These schemes provide flexibility in controlling information flow and latent space utilization.

When the rate is fixed, the optimization objective further simplifies to only the reconstruction term. Typically, increasing the rate tends to decrease the distortion (reconstruction) term. Therefore, the previously fixed global rate serves as a more interpretable hyperparameter (unit: nats/bits) to control the trade-off between the rate and the distortion term, as opposed to using a KL divergence weight $\beta$, as seen in approaches like *beta-VAE* (Higgins et al., 2016) and *DVIB* (Alemi et al., 2017), which has no interpretable meaning.

If the rate function $\delta(\mathbf{x}) = (\delta^{1/2}(\mathbf{x}))^2$ is parameterized to have a lower bound, for example $\delta(\mathbf{x}) = \delta_0 + |\tilde{\delta}(\mathbf{x})|$, it corresponds to the concept of *committed rate*, which *delta-VAE* (Razavi et al., 2019) aims to address. However, their approach is more complicated and less flexible compared to our approach.

### 4.3 Unconstrained Parameterization of a Prior Distribution

Similar to the conventional Gaussian distribution, the prior distribution can be parameterized as $(\boldsymbol{\mu}_0, \boldsymbol{\sigma}_0)$ with diagonal covariance or $(\boldsymbol{\mu}_0, \boldsymbol{\Sigma}_0^{1/2})$ with full covariance. In the previous VAE example, we observe that the prior distribution influences only the reconstruction term when generating reparameterized samples from the Slashed Normal.

As discussed earlier in Sec. 3.4, the Slashed Normal accommodates a degenerate Gaussian prior, where $\boldsymbol{\sigma}_0$ or $\boldsymbol{\Sigma}_0^{1/2}$ need not be positive or positive definite. Consequently, the actual prior parameters, $(\boldsymbol{\mu}_0, \boldsymbol{\sigma}_0)$ or $(\boldsymbol{\mu}_0, \boldsymbol{\Sigma}_0^{1/2})$, can be left unconstrained.

Let us delve into the sampling procedure for the multivariate Slashed Normal, $\mathcal{N}(\boldsymbol{\Psi}, \boldsymbol{P}, \boldsymbol{\mu}_0, \boldsymbol{\Sigma}_0)$, which is relative to a multivariate Gaussian prior $\mathcal{N}(\boldsymbol{\mu}_0, \boldsymbol{\Sigma}_0)$:

$$z = \boldsymbol{\mu}_0 + \boldsymbol{\Sigma}_0^{1/2} \underbrace{\left(\sqrt{2}\boldsymbol{a} + \boldsymbol{P}\left(stdplus(\sqrt{2}\boldsymbol{b}) \odot \epsilon\right)\right)}_{\substack{\text{sample from Slashed Normal} \\ \text{with standard Gaussian prior}}}, \text{where } \epsilon \sim \mathcal{N}(\mathbf{0}, \mathbf{1}). \tag{22}$$

This equation clearly demonstrates that a certain multivariate Gaussian prior can be implicitly incorporated by applying a linear layer or hypernetworks (Ha et al., 2017) with unconstrained weights

to samples from the Slashed Normal with a standard Gaussian prior. This property further simplifies the modeling process. This discussion also highlights the long-ignored fact that the linear projection layer on the decoder side applied on the sampled latents is effectively part of the prior distribution, which can itself be a source of collapse.

## 5  INTERPRETING THE KL AMPLITUDE

The stochastic layer formulated by the Slashed Normal Parameterization reveals interesting interpretation. We first establish the relationship between the KL amplitude and the expected gradient. For clarity, without loss of generality, we use the version of the Slashed Normal with diagonal covariance and the standard normal prior.

**Theorem 5.1** (Posterior Stationary Equation). *For the stochastic layer $z \sim \mathcal{N}(\psi)$, assuming the loss can be splitted into two terms $L(\psi) = \mathbb{E}_{z \sim \mathcal{N}(\psi)}[L_{>}(z)] + \beta D_{KL}(\mathcal{N}(\psi) || \mathcal{N}(0, 1))$, which is the case for the VAE/VIB defined via Slashed Normal, the stationary posterior distribution such that $\nabla L(\psi) = 0$ satisfies*

$$\psi = -\frac{1}{2\beta} \mathbb{E}_{\epsilon \sim \mathcal{N}(0,1)} [\nabla_{\psi} L_{>}(z = \mu + \sigma \odot \epsilon)], \qquad (23)$$

*where $\mu = \sqrt{2}\Re(\psi)$ and $\sigma = stdplus(\sqrt{2}\Im(\psi))$.*

*Proof.* Computing $\nabla L(\psi)$ and setting it to $0$ gives the result, as

$$\nabla L(\psi) = \nabla \mathbb{E}_{z \sim \mathcal{N}(\psi)}[L_{>}(z)] + 2\beta\psi = \mathbb{E}_{\epsilon \sim \mathcal{N}(0,1)}[\nabla L_{>}(z = \mu + \sigma \odot \epsilon)] + 2\beta\psi. \qquad (24)$$

$\square$

**Relationship with SmoothGrad method (Smilkov et al., 2017) for attribution**   Theorem 5.1 establishes the relationship between the locally smoothed negative gradient of $L_{>}$ with the KL amplitude $\psi$ at stationary points. This connection is reminiscent of the SmoothGrad Smilkov et al. (2017) method for attribution, which, for image classification, computes the locally smoothed gradient to obtain a clean sensitivity map identifying pixels that most affect model decisions. In this sense, SmoothGrad can be seen as performing inference for $\psi$, which is the perturbation distribution added to the input, via iterating Eq. (23) for one step. This can be viewed as an approximation for finding a rate-regularized perturbation direction. KL divergence values for specially designed bottlenecks have been directly used for attribution (Schulz et al., 2020; Jiang et al., 2020), and the gradient related to the information bottleneck has also been explored (Cheng et al., 2024). Our result connects these approaches, providing a unified perspective on attribution methods based on variational information bottlenecks and smoothed gradients.

**Implication for understanding posterior collapse**   Posterior collapse, characterized by the total or partial inactivation of latent space dimensions, is often indicated by near-zero KL divergence values. Theorem 5.1 suggests that a collapsed stationary posterior coincides with a gradient magnitude close to zero. Moreover, near stationary point, the KL divergence term can be interpreted as a penalty on the gradient magnitude. During optimization, the near-zero gradient at a certain state of collapse will make it challenging for gradient-based algorithms to escape. This is evidenced by several works (Bowman et al., 2015; Fu et al., 2019; He et al., 2019) that attempt to control the optimization trajectory to avoid being trapped in such adverse states. For mitigating posterior collapses, one can either choose to lower bound $||\psi||_2^2$, e.g, Zhu et al. (2020); Razavi et al. (2019), or the gradient norm $||\nabla_{\psi} L||$, e.g., using a Brenier map as in Wang et al. (2021); Kinoshita et al. (2023). Our result connects the two strategies at stationarity.

## 6  EXPERIMENT

### 6.1  FIXED RATE VARIATIONAL INFORMATION BOTTLENECK

Following the motivational example in the introduction, we evaluate training a VIB on MNIST and CIFAR10, directly targeting a specific rate using various normalization strategies proposed in section 4.2. In our case with fixed rate, the objective only include the cross entropy loss for

classification, and our fixed-rate VIB layer functions similarly to *dropout*. Following the setup of Alemi et al. (2017), we perform supervised classification on the MNIST and CIFAR10 datasets. We use accuracy under the Fast Gradient Sign Method (FGSM) with varying attack strengths as the metric. This choice is motivated by the known ability of VIB to improve robustness against adversarial attacks (Alemi et al., 2017).

In our experiments, we insert a fixed-rate VIB (FR-VIB) before the last linear layer preceding the final softmax layer. We test three normalization types: *batch*, *instance*, and *feature*, as proposed in section 4.2, to achieve a target average rate $\delta$ per instance. For classification with $C$ classes, where $\log C$ nats is the maximum entropy for encoding classes, we set $\delta = r \log C$. The constant $r$ is adjustable, allowing flexibility based on empirical data or theoretical insight.

Results are shown in table 1, with experimental details provided in appendix E. For both datasets, we can see that FR-VIB improves significantly against the base model on robustness against FGSM attack. Among the normalization methods, *batch* generally performs best across different values of $r$, while *instance* performs worst. We conjecture that this is due to the varying tightness of the capacity bound implied by different normalization methods. The results suggest that $r = 1$ is a good default value, aligning with the upper bound of the entropy for predicting $C$ classes. Moreover, the best error rate with $\epsilon = 0, r = 1$ in MNIST experiment is consistent with that of Alemi et al. (2017), which was obtained using a tuned value of $\beta$, suggesting the effectiveness of the proposed FR-VIB.

| MNIST | | | | | | | | | | | | | | | |
|---|---|---|---|---|---|---|---|---|---|---|---|---|---|---|
| Norm | batch | | | | | instance | | | | | feature | | | | |
| $r$ \\ $\epsilon$ | 0.0 | 0.1 | 0.2 | 0.3 | 0.4 | 0.0 | 0.1 | 0.2 | 0.3 | 0.4 | 0.0 | 0.1 | 0.2 | 0.3 | 0.4 |
| 0.125 | 1.14 | 6.46 | 13.21 | 21.72 | 35.02 | 19.28 | 24.17 | 30.27 | 37.28 | 44.13 | 1.22 | 6.93 | 17.80 | 34.60 | 52.62 |
| 0.25 | 1.19 | 6.47 | 11.17 | 16.72 | 24.42 | 5.30 | 10.61 | 15.56 | 21.69 | 28.56 | 1.27 | 6.52 | 13.60 | 23.36 | 36.03 |
| 0.5 | 1.31 | 6.71 | 15.53 | 25.28 | 35.56 | 1.62 | 5.95 | 10.12 | 15.15 | 21.69 | 1.32 | 6.62 | 11.66 | 17.46 | 25.73 |
| 1 | 1.14 | 6.24 | 10.04 | 13.20 | 17.90 | 1.36 | 5.30 | 9.82 | 16.15 | 22.87 | 1.25 | 6.19 | 9.54 | 13.90 | 20.62 |
| 1.5 | 1.19 | 6.03 | 10.77 | 15.81 | 25.90 | 1.44 | 5.49 | 10.13 | 17.15 | 24.32 | 1.21 | 6.08 | 10.44 | 14.94 | 21.66 |
| base | 1.35 | 14.94 | 58.94 | 81.52 | 89.75 | | | | | | | | | | |
| dropout | 1.20 | 10.40 | 42.45 | 70.27 | 81.55 | | | | | | | | | | |
| CIFAR10 | | | | | | | | | | | | | | | |
| Norm | batch | | | | | instance | | | | | feature | | | | |
| $r$ \\ $\epsilon$ | 0.0 | 0.1 | 0.2 | 0.3 | 0.4 | 0.0 | 0.1 | 0.2 | 0.3 | 0.4 | 0.0 | 0.1 | 0.2 | 0.3 | 0.4 |
| 0.125 | 7.28 | 58.37 | 82.06 | 88.81 | 89.64 | 25.91 | 66.90 | 84.76 | 87.79 | 88.55 | 7.03 | 57.75 | 81.51 | 87.95 | 89.14 |
| 0.25 | 7.40 | 62.09 | 86.38 | 89.39 | 89.83 | 11.69 | 56.83 | 79.10 | 83.21 | 84.24 | 7.83 | 60.31 | 77.34 | 83.30 | 86.34 |
| 0.5 | 6.94 | 59.03 | 82.24 | 86.66 | 87.49 | 7.05 | 60.04 | 83.49 | 88.12 | 89.10 | 8.80 | 56.21 | 75.73 | 82.07 | 85.51 |
| 1 | 6.42 | 48.44 | 75.90 | 86.13 | 88.09 | 6.96 | 53.86 | 76.12 | 84.87 | 87.15 | 6.82 | 51.90 | 77.46 | 85.35 | 87.18 |
| 1.5 | 6.65 | 52.83 | 79.85 | 85.70 | 87.36 | 7.14 | 51.12 | 71.15 | 80.83 | 86.36 | 6.73 | 65.87 | 83.21 | 86.64 | 87.94 |
| base | 6.70 | 91.94 | 91.29 | 90.22 | 89.83 | | | | | | | | | | |

Table 1: Error Rates on MNIST and CIFAR10: This table presents the impact of adversarial examples, generated using the Fast Gradient Sign Method (FGSM), on error rates. The values of $\epsilon$ indicate the strength of the adversarial example generated by the Fast Gradient Sign Method (FGSM). $r$ represents the predetermined KL divergence value, as a fraction of $\log C$, where $C$ is the number of classes. *batch*, *instance*, and *feature* are normalization methods used to normalize the KL divergence value.

The results demonstrate that FR-VIB can effectively control the information bottleneck without the need for a separate KL loss term or $\beta$-tuning. This simplifies the training process while maintaining or improving performance, particularly in terms of adversarial robustness. The superiority of *batch* normalization suggests that allowing some flexibility in rate allocation across the batch may be beneficial, balancing between strict per-instance control (*instance* normalization) and global per-dimension control (*feature* normalization).

## 6.2 MITIGATING POSTERIOR COLLAPSE

This experiment aims to demonstrate the versatility of Slashed Normal in addressing posterior collapse, a common issue in variational autoencoders. We benchmark various renormalization techniques and compare them with existing methods. We also tested directly adding skip connection in the hope that it will mitigate posterior collapse by mitigating gradient vanishing, as these two phenomena are closely related (see section 5).

**(Re)normalization for a target KL value** For Slashed Normal, the KL divergence takes the form of the squared $L^2$-norm of $\psi$. We test three normalization mechanisms from section 4.2: *batch*, *instance*, and *feature*, imposing the target KL value by renormalizing $\psi$ with the squared root of the target value $\tilde{\delta} = \delta_0 + |\delta|$, where $\delta_0$ is a fixed base rate and $\delta$ is learnable. We also experimented with applying renormalization only on the real part (mean) of the KL amplitude, which was done in (Zhu et al., 2020) as a special case of the proposed *feature* normalization.

**Decoupling KL divergence with batch normalization.** We also test using a learnable scalar with a large initial value instead of a fixed constant to enforce the KL divergence value. In this case, the KL divergence value is directly represented by this parameter, and we effectively decoupled learning the KL divergence value from the model architecture. We use this strrategy with *batch* normalization.

**Metrics** We evaluate using negative log-likelihood (NLL), average KL divergence, Active Units (AU) (Alemi et al., 2018), and Mutual Information $\mathrm{MI}_Q$ (Burda et al., 2015). Details are in appendix F.

**Baseline** Plain LSTM, LSTM VAE, KL warmup (Bowman et al., 2015), KL cyclic annealing (Fu et al., 2019), and BatchNorm(Zhu et al., 2020). Here we only include baselines that are applicable on the same model architecture (LSTM encoder/decoder), therefore excluding methods such as Wang et al. (2021) and Kinoshita et al. (2023). Results are in table 2.

| | NLL | KL | AU | $\mathrm{MI}_q$ |
|---|---|---|---|---|
| LSTM | 336.47 | | | |
| LSTM VAE | 337.21 | 0.00 | 0 | 0.00 |
| LSTM VAE Warmup | 336.72 | 1.09 | 1 | 1.08 |
| LSTM VAE Cyclic | 335.56 | 4.70 | 6 | 4.54 |
| Batch Mean only $\delta_0 = 6$ | 336.89 | 8.04 | 7 | 6.42 |
| Batch $\delta_0 = 6$ | 336.86 | 6.09 | 5 | 5.90 |
| Instance Mean only $\delta_0 = 6$ | 335.80 | 8.02 | 11 | 6.80 |
| Instance $\delta_0 = 6$ | 337.15 | 6.27 | 4 | 6.11 |
| Feature Mean only $\delta_0 = 6$ | 338.49 | 6.12 | 32 | 3.70 |
| Feature $\delta_0 = 6$ | 336.95 | 5.98 | 32 | 4.11 |
| BatchNorm Zhu et al. (2020) | 337.22 | 5.88 | 32 | 3.85 |
| LSTM+Skip Connection | 331.90 | 7.42 | 10 | 6.63 |
| Decoupled Learnable Rate, init $\delta = 2$ | 337.05 | 1.04 | 1 | 1.03 |
| Decoupled Learnable Rate, init $\delta = 8$ | 337.04 | 3.02 | 3 | 2.95 |
| Decoupled Learnable Rate, init $\delta = 20$ | 336.02 | 3.42 | 4 | 3.33 |
| Decoupled Learnable Rate, init $\delta = 40$ | 335.59 | 4.82 | 6 | 4.65 |
| Decoupled Learnable Rate, init $\delta = 80$ | 335.50 | 5.47 | 6 | 5.26 |

Table 2: Posterior collapse experiment.

Our results shows that:

1. Competitive Performance: Several of our methods outperform the chosen baselines (KL warmup, cyclic annealing, BatchNorm), demonstrating the effectiveness of our approach.
2. Benchmarking Renormalization Techniques: We demonstrate various ways of applying our proposed renormalization technique to the encoder's raw outputs. This reveals how different applications of renormalization affect model behavior. Certain variations, for instance, "feature" normalization ensures all latent codes are active (100% utilization), which, while not optimal for NLL, can be desirable in certain scenarios.
3. Comparison with (Zhu et al., 2020): The result on fully occupied active units (AU) clearly demonstrates the connection between Zhu et al. (2020) and the proposed *feature* normalization.
4. Simplified KL Control: By decoupling the KL divergence as an individual trainable parameter initialized with a large value (Batch Learnable Rate rows in the table), we achieve performance comparable to tuned cyclic annealing schedules. Importantly, this doesn't require scheduled modifications to the objective function, simplifying the training process.
5. Architectural Insights: The "LSTM+Skip Connection" case, which applies no specific technique to mitigate posterior collapse, outperforms all other cases. This supports our theoretical insights in Section 5 connecting posterior collapse with gradient vanishing. It suggests that model architecture may play a larger role in mitigating posterior collapse than specific tricks.

# 7 CONCLUSION

In this work, we introduced the Slashed Normal, a novel parameterization for Gaussian posterior distributions in variational inference that provides explicit control over the KL divergence via the KL amplitude. Experiments validated the effectiveness of Slashed Normal in preventing posterior collapse and enabling training information bottleneck models by directly specifying the desired KL divergence. We believe that simplicity and interpretability make the proposed parameterization a valuable addition to the toolkit for research on Variational inference based latent variable models.

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

## A  MORE ON *stdplus* FUNCTION

### A.1  DERIVATIVE OF *stdplus* FUNCTION

In this section, we derive the derivative of the proposed *stdplus* function.

For $y = stdplus(x)$, by definition we have:

$$\log(y^2) - y^2 = -x^2 - 1. \tag{25}$$

Taking the derivative w.r.t. $x$ on both sides, we have

$$\frac{2}{y}\frac{dy}{dx} - 2y\frac{dy}{dx} = -2x. \tag{26}$$

Then we obtain

$$\frac{dy}{dx} = \frac{x}{y - \frac{1}{y}}. \tag{27}$$

Both the denominator and the numerator equal 0 as $x \to 0$ as $stdplus(0) = 1$. By L'Hôpital's rule, as $x \to 0^+$ or $x \to 0^-$, we have

$$\frac{dy}{dx} = \frac{1}{2\frac{dy}{dx}}. \tag{28}$$

That is,

$$\left(\frac{dy}{dx}\bigg|_{x=0}\right)^2 = \frac{1}{2}. \tag{29}$$

It is clear that $\frac{dy}{dx} > 0$ for both sides around $x = 0$, then it gives

$$\lim_{x\to 0^-} stdplus'(x) = \lim_{x\to 0^+} stdplus'(x) = stdplus'(0) = \frac{1}{\sqrt{2}}, \tag{30}$$

which also confirms the differentiability of $stdplus(x)$.

In summary, the derivative of the proposed *stdplus* function is

$$stdplus'(x) = \begin{cases} \frac{1}{\sqrt{2}}, & x = 0 \\ \frac{stdplus(x)x}{(stdplus(x))^2 - 1}, & x \neq 0 \end{cases}. \tag{31}$$

### A.2  NUMERICAL RECIPE FOR *stdplus*($x$)

In this section, we present our numerical methods for evaluating the proposed *stdplus*($\cdot$) function, which is based on Newton's method.

From the above analysis, there is a removable discontinuity ($x = 0$) in the derivative shown in Eq. (31). Therefore, the numerical computation of *stdplus* around $x = 0$ can be inaccurate and unstable with the Newton method.

To address this, we obtain a Padé approximant of $\log stdplus$ for small $x$:

$$\log stdplus(x) \approx \frac{\frac{x}{\sqrt{2}} + \frac{x^2}{4} + \frac{x^3}{90\sqrt{2}}}{1 + \frac{5x}{6\sqrt{2}} + \frac{17x^2}{180}}, \tag{32}$$

which has an absolute error $< 3.14 \times 10^{-13}$ for $|x| < 0.04$.

For other cases ($x < 0.04$ and $x > 0.04$), we find that it suffices to use an initial guess of $\frac{1}{2}(x + \sqrt{x^2 + 4})$ (*squareplus* Barron (2021)), to allow the same Newton step to be applied for both cases of ($x < 0.04$ and $x > 0.04$). Moreover, we observe an improved numerical stability by computing $\log stdplus$ and then exponentiating to obtain stdplus.

The complete algorithm for computing $\log stdplus$ is illustrated in Algorithm 1. The update equation is inspired by the numerical methods used to evaluate the Lambert W function Lóczi (2022). In Fig. 3, we present empirical results illustrating the number of iterations used in the algorithm to achieve the desired precision. The figure indicates that 4 iterations are needed for *float32*, while *float64* requires 5 iterations.

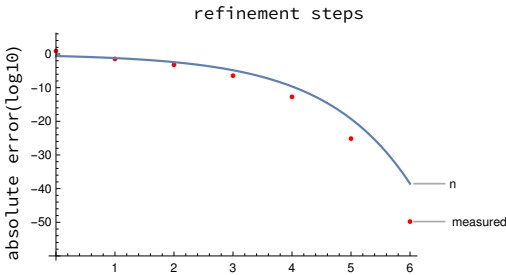

Figure 3: Number of Iterations in Algorithm 1 versus the maximum absolute error. Here, $n$ denotes the number of iterations performed in Algorithm 1 as determined by the desired precision (solid line). The term *measured* indicates the actual error.

---

**Algorithm 1** Numerical evaluation of the *stdplus* function.

---

    **function** LOG_STDPLUS($x$: input, $eps$: desired precision)
        **if** $x \in [-0.04, 0.04]$ **then**
            **return** $\dfrac{\frac{x}{\sqrt{2}} + \frac{x^2}{4} + \frac{x^3}{90\sqrt{2}}}{1 + \frac{5x}{6\sqrt{2}} + \frac{17x^2}{180}}$
        **end if**
        $r \leftarrow 2\log(\frac{1}{2}(x + \sqrt{x^2 + 4}))$              $\triangleright\, r = \log(\mathrm{stdplus}(x)^2)$
        **for** $i = 1$ to $\lceil \log_2(-\log_2(eps)) - 1 \rceil$ **do**
            $a \leftarrow \max(r, 0)$
            $r \leftarrow \dfrac{(r-1)e^{r-a} + (x^2+1)e^{-a}}{e^{r-a} - e^{-a}}$
        **end for**
        **return** $r/2$
    **end function**
    **function** STDPLUS($x$: input, $eps$: desired precision)
        **return** exp(LOG_STDPLUS(x, eps) )
    **end function**

---

## B   DERIVATION OF THE MULTIVARIATE VERSION OF SLASHED NORMAL

For the multivariate posterior distribution $\mathcal{N}(\boldsymbol{\mu}, \boldsymbol{\Sigma})$ and prior $\mathcal{N}(\boldsymbol{\mu}_0, \boldsymbol{\Sigma}_0)$, the KL divergence between them is given by

$$D_{\mathrm{KL}}(\mathcal{N}(\boldsymbol{\mu}, \boldsymbol{\Sigma}) || \mathcal{N}(\boldsymbol{\mu}_0, \boldsymbol{\Sigma}_0)) = \frac{1}{2}\left\{ \mathrm{Tr}(\boldsymbol{\Sigma}_0^{-1}\boldsymbol{\Sigma}) + (\boldsymbol{\mu} - \boldsymbol{\mu}_0)^T \boldsymbol{\Sigma}_0^{-1}(\boldsymbol{\mu} - \boldsymbol{\mu}_0) - k + \ln \frac{|\boldsymbol{\Sigma}|}{|\boldsymbol{\Sigma}_0|} \right\} \tag{33}$$

where $k$ is the dimension of the vector.

Let

$$\boldsymbol{\mu} = \boldsymbol{\mu}_0 + \boldsymbol{\Sigma}_0^{\frac{1}{2}} \boldsymbol{\mu}_\Delta$$
$$\boldsymbol{\Sigma} = (\boldsymbol{\Sigma}_0^{\frac{1}{2}}) \boldsymbol{\Sigma}_\Delta (\boldsymbol{\Sigma}_0^{\frac{1}{2}})^T, \tag{34}$$

For now, we assume that $\boldsymbol{\Sigma}_0$ and $\boldsymbol{\Sigma}_\Delta$, are full rank and $\boldsymbol{\Sigma}_0^{\frac{1}{2}}$ is a matrix such that $\boldsymbol{\Sigma}_0 = \boldsymbol{\Sigma}_0^{\frac{1}{2}}(\boldsymbol{\Sigma}_0^{\frac{1}{2}})^T$. Substituting eq. (34) into eq. (33) gives

$$D_{\mathrm{KL}} = \frac{1}{2}\left\{ \mathrm{Tr}(\boldsymbol{\Sigma}_\Delta) + \boldsymbol{\mu}_\Delta^T \boldsymbol{\mu}_\Delta - k + \log |\boldsymbol{\Sigma}_\Delta| \right\} \tag{35}$$

which only depends on the relative parameters $(\boldsymbol{\mu}_\Delta, \boldsymbol{\Sigma}_\Delta)$. To derive the multivariate version of Slashed Normal, we focus on these relative parameters. Assuming positive semidefinite, $\boldsymbol{\Sigma}_\Delta$ accepts a factorized form:

$$\boldsymbol{\Sigma}_\Delta = \boldsymbol{P}\boldsymbol{\Lambda}\boldsymbol{P}^T = \boldsymbol{P}\boldsymbol{\Lambda}^{\frac{1}{2}}(\boldsymbol{P}\boldsymbol{\Lambda}^{\frac{1}{2}})^T \tag{36}$$

where $P$ is an orthogonal matrix and $\Lambda$ is a diagonal matrix. Substituting eq. (36) into eq. (35) gives:

$$D_{\text{KL}} = \sum_{i=0}^{k-1} \frac{1}{2} \left[ (\Lambda^{1/2})_i^2 + (\boldsymbol{\mu}_\Delta)_i^2 - 1 - 2\log((\Lambda^{1/2})_i) \right] \tag{37}$$

Here, we recover the KL divergence equation of the diagonal covariance Gaussian case, which can be transformed into squared $l_2$-norm of $\boldsymbol{\psi} = \boldsymbol{a} + \boldsymbol{b}i$ by applying Slashed Normal parameterization $\boldsymbol{\psi} = \boldsymbol{a} + \boldsymbol{b}i$ that sets

$$\boldsymbol{\mu}_\Delta = \sqrt{2}\boldsymbol{a}$$
$$\Lambda^{1/2} = diag(stdplus(\sqrt{2}\boldsymbol{b})). \tag{38}$$

Combining eq. (36),eq. (38) into eq. (34) yields

$$\boldsymbol{\Sigma} = (\boldsymbol{\Sigma}_0^{\frac{1}{2}})\boldsymbol{P}\Lambda^{\frac{1}{2}}(\boldsymbol{P}\Lambda^{\frac{1}{2}})^T(\boldsymbol{\Sigma}_0^{\frac{1}{2}})^T \tag{39}$$

Finally

$$\boldsymbol{\mu} = \boldsymbol{\mu}_0 + \sqrt{2}\boldsymbol{\Sigma}_0^{1/2}\boldsymbol{a}$$
$$\boldsymbol{\Sigma}^{1/2} = \boldsymbol{\Sigma}_0^{1/2}\boldsymbol{P}diag(stdplus(\sqrt{2}\boldsymbol{b})) \tag{40}$$

We have thus recovered the multivariate Slashed Normal parameterization given in section 3.4.

**Generalization to degenerate normal distribution**    We can remove the requirement of a nondegenerate prior covariance matrix $\boldsymbol{\Sigma}_0$ by formulating the prior with the degenerate normal distribution (Mikheev, 2006; Schoeman et al., 2021).

We can conveniently express the KL divergence in this case by looking at the limit of adding a small identity matrix to the prior covariance. Note that adding $\lambda\boldsymbol{I}$ with arbitrary small $\lambda > 0$ to $\boldsymbol{\Sigma}_0$ will make it full rank, then it is obvious that:

$$\begin{aligned} &D_{\text{KL}}(\mathcal{N}(\boldsymbol{\psi}, \boldsymbol{P}, \boldsymbol{\mu}_0, \boldsymbol{\Sigma}_0)||\mathcal{N}(\boldsymbol{\mu}_0, \boldsymbol{\Sigma}_0)) \\ &= \lim_{\lambda \to 0^+} D_{\text{KL}}(\mathcal{N}(\boldsymbol{\psi}, \boldsymbol{P}, \boldsymbol{\mu}_0, \boldsymbol{\Sigma}_0 + \lambda\boldsymbol{I})||\mathcal{N}(\boldsymbol{\mu}_0, \boldsymbol{\Sigma}_0 + \lambda\boldsymbol{I})) \\ &= \boldsymbol{\psi}^H\boldsymbol{\psi} \end{aligned} \tag{41}$$

This result highlights the property that the KL divergence for Slashed Normal is independent of the prior distribution, even in the degenerate case.

## C    PROOF FOR THEOREM 4.1

$$\begin{aligned} I(X; Z) &= \mathbb{E}_{\mathbf{x}}\mathbb{E}_{\mathbf{z} \sim q(\mathbf{z}|\mathbf{x})}[\log \frac{q(\mathbf{z}|\mathbf{x})}{q(\mathbf{z})}] \\ &= \mathbb{E}_{\mathbf{x}}\mathbb{E}_{\mathbf{z} \sim q(\mathbf{z}|\mathbf{x})}[\log \frac{q(\mathbf{z}|\mathbf{x})}{p(\mathbf{z})}] - D_{KL}(q(\mathbf{z})||p(\mathbf{z})) \\ &\leq \mathbb{E}_{\mathbf{x}}\mathbb{E}_{\mathbf{z} \sim q(\mathbf{z}|\mathbf{x})}[\log \frac{q(\mathbf{z}|\mathbf{x})}{p(\mathbf{z})}] \\ &= \mathbb{E}_{\mathbf{x}}D_{KL}(q(\mathbf{z}|\mathbf{x})||p(\mathbf{z})) \\ &= \mathbb{E}_x||\boldsymbol{\psi}||_2^2 = \text{Channel Capacity,} \end{aligned} \tag{42}$$

where the equality is achieved when $D_{KL}(q(\mathbf{z})||p(\mathbf{z})) = 0$.

## D    COMPUTATIONAL RESOURCES

All experiments reported in this paper were performed on a server equipped with an NVIDIA GeForce RTX 3090 GPU and 64GB of RAM.

# E  EXPERIMENT DETAILS ON FIXED RATE VARIATIONAL INFORMATION BOTTLENECK

## E.1  OVERVIEW

**Motivation**  Existing IB-based approaches, such as the deep variational information bottleneck (VIB) Alemi et al. (2017) and $\beta$-VAE Higgins et al. (2016), use a hyperparameter $\beta$ (e.g., in eqn 1) to control the compression strength for the encoded representation. However, in practice, we find that tuning $\beta$ is quite tricky for the following reasons: 1. different tasks and model architectures may require different $\beta$ values that differ in several magnitudes, requiring extensive experimentation to identify; 2. certain range of $\beta$ may make the training process vulnerable to the phenomenon of posterior collapse, making the training process unstable; 3. it increases the complexity of balancing different loss terms when multiple loss terms are present.

**FR-VIB**  In response to these challenges, we propose a variant of the variational information bottleneck, termed the Fixed-Rate Variational Information Bottleneck (FR-VIB). This approach specifies the KL divergence directly as a hyperparameter, circumventing the indirect control mechanisms associated with $\beta$. The component is formalized as:

$$\mathbf{z} \sim \mathcal{N}(\mathbf{z}; \boldsymbol{\psi}(\mathbf{x})), \quad s.t. \quad \mathbb{E}_{\mathbf{x}}[||\boldsymbol{\psi}||_2^2] = \delta \tag{43}$$

where $\delta$ is the predetermined kl divergence (rate) value.

**Training Objective**  The training objective is defined as:

$$\min_{\theta} \quad \mathbb{E}_{\mathbf{x} \sim p_{\text{data}}(\mathbf{x})} \mathbb{E}_{\mathbf{z} \sim \mathcal{N}(\mathbf{z}; \boldsymbol{\psi}_{\theta}(\mathbf{x}))}[-\log p_{\theta}(\mathbf{y}|\mathbf{z})]$$
$$s.t. \quad \mathbb{E}_{\mathbf{x}}[||\boldsymbol{\psi}||_2^2] = \delta \tag{44}$$

where $\mathbf{y}$ denotes the label in a multiclass classification setting. The constraint here is enforced at a parameterization level through the strategies introduced in sec.4.2 by controlling the $L^2$-norm of the KL amplitude vector.

**Normalization implementations**  As discussed in Section 4.2, we employ three normalization strategies, namely batch, instance, and feature normalization, to achieve the desired KL divergence. We refer to these three ways of normalization as *batch*, *instance*, and *feature* normalization. Batch and feature normalization utilize mini-batch statistics during training; and, at the test time, running statistics updated during training are used for normalization, which is similar to *BatchNorm* Ioffe and Szegedy (2015). Instance normalization directly applies $L^2$ normalization to each $\psi(\mathbf{x})$.

## E.2  DATASETS

We tested the proposed *FR-VIB* on the task of multiclass classification on *MNIST* and *CIFAR10* datasets. For both datasets, this bottleneck layer is placed before the last linear projection. All images are scaled to have pixel values between $-1$ and $1$.

**MNIST**  We follow the model architecture as in Alemi et al. (2017), which is structured as a multilayer perceptron (MLP) with layers configured as 784-1024-1024-512-10 and employing ReLU activation functions, We treat the 512-sized output as the raw KL amplitude vector $\tilde{\psi}$, which is a complex vector of 256 dimensions. This vector undergoes renormalization to meet the desired $L^2$ norm. We use Adam optimizer Kingma and Ba (2014) with an initial learning rate of $1\mathrm{e}{-4}$ that decays by a factor of 0.99 every 2 epoches; weight decay $1\mathrm{e}{-4}$. Models are trained for 400 epochs. Following Alemi et al. (2017), we take the average from 12 posterior samples to make a prediction during the evaluation. The baseline model is the same architecture with the bottleneck layer removed (*base*). We also trained the same baseline, but with dropout rate 0.2 (*drop*).

**CIFAR10**  The setup for CIFAR10 closely follows that of MNIST, except that we use *Resnet18* from *torchvision* maintainers and contributors (2016), and the output layer has a dimension of 512; the initial learning rate is set to $2\mathrm{e}{-4}$ which decays by a factor of 0.98 for every 2 epochs.

### E.3 THE FGSM METHOD

The adversarial examples are generated by the Fast Gradient Sign Method (FGSM) Goodfellow et al. (2015), where the attack example is generated by

$$\tilde{x} = x + \epsilon \cdot sign(\nabla_x L(\theta, x, y)), \tag{45}$$

where $L(\theta, x, y)$ represents the cross-entropy loss for the data $x$ with label $y$. For both datasets, we can see that FR-VIB improves significantly against the base model on robustness against adversarial examples.

## F EXPERIMENT DETAILS ON POSTERIOR COLLAPSE EXPERIMENT

### F.1 EVALUATION METRICS

**(mean) KL divergence (KL)**

$$KL = \mathbb{E}_{p_{\text{data}}(\boldsymbol{x})}[D_{KL}(q_(\boldsymbol{z}|\boldsymbol{x})||p(\boldsymbol{z}))] \tag{46}$$

**Active Unit (AU) (Burda et al., 2015)**  This metric is defined as the number of latent dimensions that are active. The activation of latents is defined as

$$AU = Cov(\mathbb{E}_{\boldsymbol{z} \sim q(\boldsymbol{z}|\boldsymbol{x})}[\boldsymbol{z}]) \tag{47}$$

We follow the convention that a dimension $i$ is active if $AU_i > 0.01$.

**Mutual information $I_q$ (Alemi et al., 2017)**

$$I_q = \mathbb{E}_{p_{\text{data}}(\boldsymbol{x})}[D_{KL}(q(\boldsymbol{z}|\boldsymbol{x})||p(\boldsymbol{z}))] - D_{KL}(q(\boldsymbol{z})||p(\boldsymbol{z})) \tag{48}$$

where $p_{\text{data}}(\boldsymbol{x})$ is the data distribution. $q(\boldsymbol{z}) = \mathbb{E}_{\boldsymbol{x} \sim p_{\text{data}}(\boldsymbol{x})}q(\boldsymbol{z}|\boldsymbol{x})$ is the marginal distribution of $\boldsymbol{z}$. $p(\boldsymbol{z})$ is the prior for $\boldsymbol{z}$. This metric measures how much information content about $\boldsymbol{x}$ is encoded in $\boldsymbol{z}$. When the second term is small (the amortization gap), the KL metric defined previously approximates this value.

### F.2 CONFIGURATION

For both encoder and decoder, we use 3 layers of LSTM with 512 hidden units. The decoder uses a dropout rate 20% between layers. We use latent dimension of 32, word embedding size 512. For estimating NLL, we use importance weighted ELBO Burda et al. (2015) using 100 samples. Training is performed for 400 epochs using the OneCycle learning rate schedule with warm-up steps of 10%.

