# OpenReview forum: "Slashed Normal: Parameterize Normal Posterior Distributions with KL Amplitude"
_ICLR.cc/2025/Conference — ICLR 2025 Conference Withdrawn Submission_

### Official Review · Reviewer_bxrV · 2024-11-02

**Soundness:** 2
**Presentation:** 2
**Contribution:** 2
**Rating:** 3
**Confidence:** 4

**Summary:**

This paper proposes a new activation function, "```stdplus```," as a replacement for conventional ```exp``` or ```softplus``` parameterization of the approximate posterior variance in Gaussian VAEs, resulting in a new distribution they call "Slashed Normal." This formulation allows for direct control over the channel capacity or information rate in VAEs and provides a more interpretable trade-off between the rate (KL) and distortion (reconstruction) terms. However, there are critical weaknesses that undermine the paper's contributions.

**Strengths:**

The authors identify several known issues within the VAE literature, such as posterior collapse and numerical instability, and aim to address them through their proposed parameterization approach. While theoretically well-motivated, the paper falls significantly short in providing sufficient empirical evaluation and validation of these claims, as discussed below.

**Weaknesses:**

A major weakness of this paper is the lack of empirical support for its primary claims. Since the main contribution centers on replacing the traditional ```exp``` or ```softplus``` parameterizations with the proposed ```stdplus``` activation, the most crucial empirical evidence should be a comprehensive evaluation of these parameterization choices across various datasets and architectures, with other factors controlled. Instead, the experiments primarily explore the impact of adversarial examples on performance, and their dependence on normalization choices, which seems tangential to the core contribution. The absence of a direct comparison between ```stdplus```, ```exp```, and ```softplus``` raises significant doubts about the practical value of the proposed method.

Additionally, it is known among practitioners that while ```exp``` is more challenging to train and may require techniques like clamping, it generally yields better performance compared to ```softplus```. This is likely attributed to the "expansive" nature of the ```exp``` nonlinearity, contrasted with the "almost linear" behavior of softplus, making the latter less expressive. Given the close relationship between the ```stdplus``` and ```softplus``` (Fig. 2b), it raises concerns that ```stdplus``` might underperform compared to ```exp``` in practical settings. Without demonstrating that ```stdplus``` is at least on par with ```exp``` or ```softplus``` in terms of empirical performance, the findings of this paper hold limited practical relevance.

Further complicating the evaluation, the paper relies on unvalidated assertions of numerical stability improvements. The authors assert (lines 309-311) that their approach "eliminates all potentially unstable operations, e.g., log/exp, which previously require clipping the range of the input to prevent numerical problems. This property likely improves the numerical stability of training." This is indeed a major challenge in training VAEs, particularly in hierarchical settings. However, without an experimental demonstration to substantiate this claim, the impact remains speculative. For a novel parameterization technique, empirical validation of stability is essential, and its absence limits the trust in ```stdplus``` as a robust alternative.

Related to this, the introduction of the ```stdplus``` function adds significant implementation complexity without sufficient justification in terms of demonstrated performance gains. As presented in Algorithm 1, ```stdplus``` is computationally more complex than a simple ```exp``` or ```softplus``` functional call. The authors need to justify this added complexity with clear, consistent performance improvements across practical applications. Yet, the current manuscript fails to establish this, leaving the reader questioning whether ```stdplus``` offers tangible benefits to warrant its more intricate setup.

While the authors acknowledge the need for more extensive empirical comparisons, this does not excuse the lack of rigorous evaluation in the current manuscript. Given the main contribution of the paper is replacing ```exp```/```softplus``` with ```stdplus```, a lack of empirical comparison between these parametrization choices almost seems like an intentionally left-out comparison.

Overall, I am inclined towards rejection. Without sufficient empirical evidence, the theoretical contributions alone are not enough to warrant publication at this venue.

**Questions:**

- What is $q(z)$ in Theorem 4.1? It is used without a definition. Is it related to the concept of "aggregated posterior" ([Chen et al., 2018](https://arxiv.org/abs/1802.04942)) or "average encoding distribution" ([Hoffman and Johnson, 2016](https://www.cs.columbia.edu/~blei/fogm/2020F/readings/HoffmanJohnson2016.pdf))?

- Line 230: Why refer to $\psi$ as the KL amplitude? It is simply a complex number. Wouldn’t the amplitude be $|\psi|$ instead?

- The writing is mostly clear but could be enhanced for better clarity. For example, the transition to the "half moon classification" example in the introduction feels abrupt, lacks proper motivation, and is highly specific. This leaves the reader puzzled about its relevance to the introduction. Furthermore, this example is not revisited later, making it appear like an irrelevant addition to the introduction. Can the authors clarify the significance of this example?

---

### Official Review · Reviewer_TLQC · 2024-11-03

**Soundness:** 2
**Presentation:** 2
**Contribution:** 2
**Rating:** 3
**Confidence:** 4

**Summary:**

The paper introduces Slashed Normal, a novel parameterization of Gaussian posterior distributions in variational-inference-based latent variable models, particularly focusing on Variational Autoencoders (VAEs). The method replaces traditional activation functions like softplus or exponential with stdplus to derive the standard deviation. By establishing a direct connection between the squared L2-norm of the raw neural network output (termed KL amplitude) and the exact KL divergence between the prior and posterior, the authors aim to provide explicit control over the KL divergence during training. They claim that this approach offers theoretical insights, enhances numerical stability, mitigates posterior collapse, and simplifies the training process.

**Strengths:**

- The method allows explicit manipulation of the KL divergence term by directly linking it to the network's output, potentially aiding in balancing the trade-off between reconstruction and regularization in VAEs.
- By controlling the KL divergence explicitly, the approach offers a potential solution to posterior collapse, a common issue where the model ignores the latent variables.
- The reformulation of the VAE loss function eliminates unstable operations like log and exp, which may improve numerical stability.

**Weaknesses:**

- The derivation of the Slashed Normal parameterization is convoluted, lacks sufficient explanation, and contains too many abuses of notation. For instance, the transition from Equation (9) to the introduction of complex numbers is abrupt and may confuse readers unfamiliar with the application of complex numbers in this context. The use of the Lambert W function is mentioned but not adequately justified or explained, making it difficult to follow the mathematical reasoning.
- In Section 2, the authors create confusion by using the term "posterior" where they should more accurately refer to the "approximate posterior."
- The experimental results are minimal and lack depth. In Section 6, while the authors mention outperforming certain baselines, they do not provide comprehensive quantitative comparisons or statistical significance tests.
- The paper acknowledges existing techniques for controlling KL divergence and mitigating posterior collapse but does not thoroughly compare the proposed method against these alternatives.
- Despite citing numerical instability as a motivation, the paper does not present empirical evidence demonstrating improved stability during training. The claim that the method "likely improves the numerical stability of training" is speculative without supporting experiments.
- The discussion on interpreting the KL amplitude and its relationship with posterior collapse is superficial. The connection made via Theorem 5.1 is not deeply analyzed, and the practical significance of this relationship is not convincingly established.

**Questions:**

- What is the rationale behind representing the KL amplitude as a complex number? Are there empirical results showing that this complex parameterization yields better performance or insights compared to a purely real-valued approach?
- The paper claims improved numerical stability due to the elimination of operations like log and exp. Can the authors provide experimental results demonstrating reduced training instability or better convergence properties compared to traditional methods?
- How does Slashed Normal perform against more recent and advanced techniques for preventing posterior collapse, such as those employing sophisticated architectures or alternative regularization methods?
- Does the introduction of complex numbers and the stdplus function introduce computational overhead or require specialized implementation?
- The authors mention different normalization strategies but do not provide practical guidelines on selecting the appropriate one. Under what circumstances should a practitioner choose batch normalization over instance or feature normalization?
- The parameterization is developed for Gaussian priors. Can the method be extended to non-Gaussian priors or to models where the posterior is not Gaussian? If not, this limits the applicability of the approach.
- While the paper discusses the KL amplitude's theoretical interpretation, how does this translate to practical benefits? Can the authors provide examples or case studies where understanding the KL amplitude leads to improved model performance or insights?

---

### Official Review · Reviewer_73p3 · 2024-11-04

**Soundness:** 3
**Presentation:** 3
**Contribution:** 3
**Rating:** 6
**Confidence:** 5

**Summary:**

The paper proposed a Slashed Normal prior that parametrizes the KL divergence term in VAE as the form of a $L^2$-norm. It enables direct control of the KL divergence. Theoretical and experimental results show that the proposed approach is able to mitigate the issue of  posterior collapse.

**Strengths:**

* The presentation and the logic flow are clear.
* The proposed method is intuitive.
* The method derivation is good, with clear math notations and solid theorem prooves.

**Weaknesses:**

* The soundness is a bit questionable. There is no code uploaded.
* The experimental results are a bit weak. For example, in experiment 1, which is the standard VAE results. There are actually two versions of standard VAE, one is the traditional KL term and the other is the reparametrized KL term. Will the results be significantly different?
* There are no error bars in both of the experiments. For example, in experiment 2, I can see that the KL terms are significantly different (which is clear and intuitive). But the NLL terms (if that is the reconstruction loss) are roughly the same. Do these results show significant/effective performance differences? Some qualitative comparison will be better.
* There is no comparison with alternative methods that also mitigate the posterior collapse issue. For example, https://proceedings.neurips.cc/paper/2017/hash/35464c848f410e55a13bb9d78e7fddd0-Abstract.html, https://proceedings.mlr.press/v161/jerfel21a.html, https://openreview.net/pdf?id=HD5Y7M8Xdk.

**Questions:**

/

---

### Official Review · Reviewer_85CT · 2024-11-04

**Soundness:** 3
**Presentation:** 2
**Contribution:** 2
**Rating:** 3
**Confidence:** 4

**Summary:**

The paper proposes a new parameterization of Gaussian variational distributions when using variational inference (VI) in probabilistic models with Gaussian priors (the discussion and empirical evaluation focuses specifically on _amortized_ VI, i.e., variational autoencoders and variational information bottleneck). The KL-divergence from a Gaussian prior to a Gaussian variational distribution can be written as a sum $a^2 + b^2$ where $a$ depends only on the mean and $b$ depends only on the variance of the variational distribution. The authors propose to parameterize the variational distribution by $a$ and $b$ (rather than by, e.g., its mean and variance, or by its natural parameters). Solving for $a$ and $b$ results in $a$ being the shift between prior to variational mean, measured in units of the prior standard deviation, while $b$ is a more complicated function of the fraction between prior and variational standard deviation.

The paper claims that the proposed parameterization, in which the KL-term in the ELBO (the "rate") takes the simple form $a^2 + b^2$, allows for easier control of the rate and helps mitigating posterior collapse.

**Strengths:**

- The paper addresses a relevant problem that might sometimes be overlooked as a technical detail.
- It discusses important consequences such as adversarial robustness and posterior collapse of the proposed method.
- I think a streamlined derivation could motivate the proposed parameterization in a very straight-forward way whose simplicity would warrant exploring it in practical applications even if there may be limited strict theoretical guarantees.

**Weaknesses:**

While the studied problems are important, the derivations seem to be correct, and there are some (limited) empirical results, I find the paper lacking both in content and in presentation.

## Content

The paper proposes a very simple (see "presentation" below) parameterization of the variational distribution in a specific model class.
In my experience, it is common when implementing probabilistic models that one thinks a bit about reasonable parameterizations of the probability distributions that avoid exploding gradients and that allow for easy regularization, initialization, and/or plotting of desired quantities.
Such considerations often make it into the appendix of a publication, where one describes details of the model implementation.
For such considerations to be noteworthy enough to merit a dedicated paper, in my opinion, they have to (i) apply to a general class of problems and (ii) be thoroughly evaluated empirically across a wide range of models to make sure that the improvements on a particular model are not an artifact of, e.g., the inevitably different initialization that comes with every reparameterization.
I find the paper to be lacking in both (i) and (ii).

**Regarding generality (i),** the proposal is limited to models with a Gaussian prior and Gaussian variational distribution.
- While this simple setup is admittedly often used in practice, the paper seems to restrict the discussion and evaluation even further to *fixed* priors.
  However, it seems to me that a good parameterization of a variational distribution would be of particularly interest in models with learned priors (which appear naturally in hierarchical VAEs [1-3], and also in applications of VAEs to data compression [4]).
  I would find it an interesting question whether a parameterization that is relative to the prior is beneficial or detrimental to optimization speed when the prior itself changes during training.
- Beyond learned priors, the idea of parameterizing the variational distribution in such a way that the rate term takes a simple form seems quite general to me, and it seems like this concept should, in some form, also be applicable to other distributions than Gaussians.

**Regarding empirical evaluations (ii),** I find the experiments somewhat limited, but this may in parts be because I did not fully understand what the baselines are.
- From the discussion, it is unclear to me whether baselines include a thorough comparison to standard $\beta$-VAEs.
  The discussion seems to suggest that the proposed family of renormalization methods do not need a tuning parameter (akin to $\beta$) because the target rate can be set directly.
  But of course, the target rate $r$ then takes the role of a tuning parameter.
  For a full comparison, I would have expected some rate/performance plot, where performance can be any of the evaluated performance metrics (e.g., adversarial robustness or NLL), and the rate is always _measured_ by the standard KL-divergence and just _controlled_ differently (either explicitly by $r$ or implicitly by $\beta$).
- Point 4 in Section 6.2 suggests that the proposed method makes it easier to control the KL-term even when its value is trained.
  However, it seems like model performance (e.g., number of active units) depends strongly on the initialization of $\delta$.
  Since the final value of the KL-term differs strongly from the initialization (see Table 2), it actually seems to me that the KL-divergence is quite hard to control in this setup.
  We usually try to find setups where final model performance does _not_ depend strongly on initialization, since the effect of different initializations on final model performance is indirect and depends in complicated ways on learning rates and the number of training iterations.
  I would imagine that it would have been much easier to control the KL-divergence had we just used a traditional parameterization of $q$ and added a simple regularization term $\propto (D_\text{KL} - \delta)^2$ to the training objective (where $\delta$ is the target rate).

I would find the limited empirical evaluation less concerning if there was clear theoretical evidence of its benefits.
However, I find the theoretical arguments somewhat vague.
For example, in the paragraph below Eq. 19, the paper highlights that the KL-divergence takes a very simple form in the proposed parameterization, claiming that "this formulation eliminates all potentially unstable operations, e.g., log/exp".
But first, other parameterizations that are common in practice avoid this too (e.g., parameterizing the variance by a softplus function).
And second, and more importantly, the claim in the paper ignores the fact that the proposed parameterization just shoves the complexity (and potential instability?) from the KL-term into the reconstruction term.

## Presentation

My main concern with the presentation is that the paper seems to overstate complexity at many points.
This is not a criticism of the simplicity of the proposal—simplicity is a good thing.
But, at several places, the paper makes simple (and sometimes even trivial) points seem unnecessarily complicated.
Examples include:

- Most importantly, a lot of space of the paper is used to derive the proposed parameterization, making it appear like this is a complicated invention that takes a lot of insight.
  I think this complexity is artificial since the result almost falls out immediately from the expression for the KL-divergence between two normal distributions (Eq. 3).
  The KL-divergence is a sum of a term that only involves the variational mean $\mu$ and a term that only involves the variational standard deviation $\sigma$.
  Why not just define these two terms as $a^2$ and $b^2$, respectively, and then solve for $\mu(a)$ and $\sigma(b)$?
  Here, $\mu(a)$ is trivial and $\sigma(b)$ involves a special function that we can't avoid anyway.
  Instead of such a simple two-line derivation, the paper first proposes a _different_ parameterization in Section 3.1, that (i) seems less well motivated to me than my above simple motivation of the eventually proposed "$a^2 + b^2$" parameterization, (ii) is derived in such detail that I found it easier to rederive it myself than to follow every algebraic step in the paper, and, most importantly, (iii) gets discarded at the end of the section anyway.
- The argument to discard the parameterization of Section 3.1 could have been seen without the lengthy derivation: if the argument is that $\frac{\partial\sigma^2}{\partial\delta} \xrightarrow{\delta\to0} \infty$, then this can be seen simply by observing that $\frac{\partial\sigma^2}{\partial\delta}$ = $1 \big/ \frac{\partial\delta}{\partial\sigma^2}$, where $\left. \frac{\partial\delta}{\partial\sigma^2} \right|_{\delta=0}=0$ since $\delta$ is the KL-divergence, so the only place where it is zero is when prior and variational distribution are equal, in which case the derivative w.r.t. $\sigma^2$ is trivially zero from Eq. 3.
- For both Theorems 4.1 and 5.1, it seems like an overstatement to me to present these as "Theorems". Theorem 4.1 is a well-known information-theoretical bound, and Theorem 5.1 just states that $\nabla_x (f(x) + x^2) = 0 \Longleftrightarrow x = -\frac{1}{2} \nabla_x f(x)$.

## Minor Point / Potential Typo

- Line 522: "Batch Learneable Rate" --> "Decoupled Learneable Rate"?

## References

- [1] [Vahdat and Kautz, NVAE: A Deep Hierarchical Variational Autoencoder, NeurIPS 2020](https://proceedings.neurips.cc/paper/2020/hash/e3b21256183cf7c2c7a66be163579d37-Abstract.html)
- [2] [Child, Very Deep VAEs Generalize Autoregressive Models and Can Outperform Them on Images, ICLR 2021](https://openreview.net/forum?id=RLRXCV6DbEJ)
- [3] [Xiao and Bamler, Trading Information between Latents in Hierarchical Variational Autoencoders, ICLR 2023](https://openreview.net/forum?id=eWtMdr6yCmL)
- [4] [Ballé et al., End-to-end Optimized Image Compression, ICLR 2017](https://openreview.net/forum?id=rJxdQ3jeg)

**Questions:**

- Is my proposed simple two-line derivation of the "$D_\text{KL} = a^2 + b^2$" parameterization correct or did I miss anything that would warrant the much more elaborate derivation in the paper?
- Can you say something about your proposal in the context of learned priors?
- Can your proposal be extended to other distributions than Gaussians? Maybe exponential family distributions?
- How does your method as a function of $r$ compare empirically to $\beta$-VAE / VIB as a function of $\beta$?
- (minor point: The discussion and evaluations seem to focus on _amortized_ variational inference. Is the method limited to amortized VI? It doesn't seem to be from a theoretical point of view, but what differences would you expect in its application to amortized vs. non-amortized VI?)

---

### Note · Authors · 2024-12-03

**Comment:**

Dear Area Chair and Reviewers,
Thank you for your thorough and constructive feedback on our manuscript. After careful consideration of the reviews and feedback received, we have decided to withdraw our submission.
We sincerely appreciate the time and effort the reviewers invested in providing detailed comments and suggestions. Your feedback will be valuable for improving our work.
Best regards,
The Authors

**Withdrawal Confirmation:**

I have read and agree with the venue's withdrawal policy on behalf of myself and my co-authors.